# Harnessing the Challenges and Solutions to Improve Security Warnings: A Review

**DOI:** 10.3390/s21217313

**Published:** 2021-11-03

**Authors:** Zarul Fitri Zaaba, Christine Lim Xin Yi, Ammar Amran, Mohd Adib Omar

**Affiliations:** School of Computer Sciences, Universiti Sains Malaysia, George Town 11800, Pulau Pinang, Malaysia; christinelimxinyi@student.usm.my (C.L.X.Y.); ammar.ucom13@student.usm.my (A.A.); adib@usm.my (M.A.O.)

**Keywords:** security warning, usability, usable security, warning timeline, warning classifications

## Abstract

The security warning is a representation of communication that is used to warn and to inform a person whether security menaces have been discovered in order to prevent any consequences of damage from taking place. The purpose of a security warning is to provide a legitimate alert (to notify and to warn) to the users so that a secure manner of action is safely conducted. It is worth noting that the majority of computer users prefer to dismiss security warnings due to lack of attention, the use of technical words, and the deficiency of information provided. This paper determines to achieve two outcomes: firstly, a thorough review of problems, challenges, and approaches to improving security warnings. Our work complements the previous classifications in the identification of problems and challenges in security warnings by value-adding a new classification, namely immersion in the primary task. Then, we add other related works within the known classifications accordingly. In addition, our work also presents the classifications of approaches to improving security warnings. Secondly, we propose two timelines by addressing the problems, challenges, and approaches to improving security warnings. It is expected that the outcomes of this research will be useful to researchers within the niche area for analysing trends and providing the groundwork in security warning studies, respectively.

## 1. Introduction

Currently, society has become very dependent on technology. All information can be accessed anywhere and anytime from the Internet. In this respect, an online survey was chosen as one of the research tools to gather the information based on the end user’s perspective. In addition, the online survey used in this research also makes it easy for the respondents to respond, considering that the usage of the Internet is very high [1]. As the usage of the Internet becomes higher, the number of security threats and risks also increases, considering the potential threats coming along with this [2]. Thus, there is a growth in the population of threats, and once the threats evolve, the risks become higher and higher. A statistic presented in [3] shows the top 10 countries suffering from the problems or vulnerability associated with cyber-attacks in their computer systems. According to [4], the threats are worldwide in various continents, and 19.8% of computers around the world have been infected by malware. Based on the given statistic, all these threats could affect the computer system, and thereby the users tend to encounter the loss of valuable assets including banking information and user credentials data. Amongst the popular and recent attacks are those derived from banking malware. These also involve malicious programs for automated teller machines (ATM) and point of sale (POS) terminals. The statistics from Kaspersky on banking malware indicate that Kaspersky can block efforts to launch one or more malicious banking malware programs that steal bank accounts [4].

Today, security warnings have been widely used to notify of any possible menaces that have been detected so that actions can be taken securely. Specifically, a security warning is the warning system (of a biological or any technical nature) developed to remind the general public about future potential malicious activities such as natural disaster warnings, road safety warnings, and food product warnings. From the perspective of computing, a security warning is a form of communication that alerts the users of possible attacks or any security violations. One of the characteristics of the security warning is that it defends the user and their computer system from any menaces to reduce the risk of security threats [5].

In one context, warnings not only provide information about the benefits and physical warning problems [6] but also act as an alert system that protects the computer systems from many threats or menaces, i.e., malware, information theft, and spoofing. In general, security warnings can be grouped into five different types, i.e., dialog box systems, in-place systems, notification systems, balloon systems, and banner systems [7,8]. It can be noted that the dialog box is one of the most ordinarily used and presented to convey to the user useful information about possible occurrences in the context of computer warnings, i.e., confidentiality, the integrity of systems, availability of information, and other valuable assets of data. 

In this paper, evidence from various scholars was gathered and presented accordingly. Most of the issues and challenges in security warnings are mentioned independently in many publications such as the technical words, the motivation for heeding the warnings, and the evaluation of the risk of warnings [9,10,11,12,13,14,15,16,17,18,19]. Most of the researchers within this domain are focusing on some specific challenges and solutions. To our knowledge, the researchers may lack detail in classifying the challenges, problems, and solutions accordingly, which this work tries to address, attempting to bridge the gaps. It is worth highlighting that [16,20] can be considered amongst the first groups of researchers to gather and classify the problems and challenges, respectively. 

There are two main outcomes of this research work: firstly, a comprehensive review of problems, challenges, and approaches to improving security warnings. Our work compliments works by [16,20] by value-adding another new classification, namely immersion in the primary task. Then, we address some other related works within the known classifications. Secondly, timelines of problems, challenges, and solutions in security warnings are proposed.

Accordingly, this paper is organized as follows: Section 2 reviews the security warnings background; Section 3 describes the problems and challenges within the domain; Section 4 presents approaches to improving security warnings; Section 5 proposes timelines of problems and challenges and approaches to improving security warnings; Section 6 presents a discussion; and Section 7 presents the conclusion coupled with the future work of this research paper.

## 2. Security Warnings Background

According to [6,7], a warning can be defined as a class of communication implemented to defend people from various dangerous occurrences, i.e., health problems, any injuries, and accidents. It also is viewed as a form of giving information to the user about any potential threats or problems that would probably occur and to protect users from any harm. In a computing context, the warning can be seen as a communication medium or channel to inform the users about any possible attacks or issues that occurred in the computer system. Microsoft [21] describes a warning as the risk or potential of losing a valuable asset such as finances, personal information, system integrity, privacy, and a user’s time. 

Amongst the key features of the security warning is that it provides a defence mechanism for the computer system from various menaces or computer threats and helps the user to mitigate the risk of becoming the victim of threats [5,22,23,24,25,26,27,28]. On the other hand, the security warning has also been viewed as an instant medium that highlights the security breaches to the users. According to [11], security warnings cover a huge scope that includes providing the benefits and challenges of the physical warning to the basic computer system. It normally informs one of any suspicious activities regarding the threats or attack that harms the computer system [29].

Various types of security warnings can be derived from one’s computer, such as an operating system defending a user’s personal computer from harm. Some of the warnings might disturb the user’s primary task, and, in another scenario, they might just appear for a while. In [21], warnings are classified into five different user interface contexts, which include a dialog box, in place, notifications, balloons, and banners, to alert users accordingly. 

## 3. Problems and Challenges of Security Warnings

A security warning is an essential medium to inform people about potential threats to avoid undesirable consequences. However, it is a must to take note that, despite the importance of the security warning, the end users still encounter significant difficulties with this. The following sub-sections highlight the end users’ experiences whilst encountering security warnings in different contexts. Previous research showed that the majority of the users do not pay attention to the risk [30,31], that they did not read the security warning text [10,32], that there are problems with the technical words [6,10,23,33], the user’s motivations towards heeding security warnings [34,35], the user’s assessment of the implication of warnings [10,36], a poor mental model [6,36,37], and that users become habituated to the security warning [38,39].

### 3.1. Lack of Understanding towards Technical Wordings

Ref. [40] conducted a study to explore the issues regarding the secure socket layer (SSL) certificates warning and the usable security of the warning. The end user’s understanding of the SSL certificates warning was examined. Accordingly, several participants claimed that they do not understand the jargon used in the warning, especially the term “encrypted”, based on the evaluation results.

An online study was conducted in [41] to evaluate the effectiveness of warning messages extracted from current browsers. Six out of twenty-eight warning messages (in a different randomized order) were presented for each participant. Based on the results, warning messages that included complex technical terms could negatively impact the users’ perceptions of a message. All participants revealed that technical terms used in the warning would hinder the understanding and awareness of potential problems.

### 3.2. Inattention towards Warnings

An eye-tracking study was carried out in [42] to investigate how much time the users spent gazing at the security indicator cues. Their findings implied that many participants did not take the warnings seriously. With this, the participants made comments such as “I did not even think to look up to the security indicator”. 

Ref. [43] conducted NeuroIS studies to gain a better perception regarding the users’ reactions to security warnings that could subsequently help them create an effective message. It has been reported in this study that the key factor affecting security behaviour is inattentiveness towards the security message. For instance, users incline to dismiss or neglect the security warning message in response to repeated warning exposure.

### 3.3. Lack of Understanding towards Warning

Ref. [44] conducted research interviews to understand users’ decision processes when encountering phishing emails (with cues). Such cues include address spoofing, secure site icons, and/or broken images on the web page. It is noteworthy that the participants might use these cues to determine whether an email or the website can be trustable, but they are unable to properly interpret the cues, unfortunately. For example, few participants could understand the potential danger (‘Not Secure’) of locked contents on a web page, but they tend to ignore the secure site lock in Chrome browsers.

Amongst the most essential issues in the security warning context is the usage of the terminologies. According to the survey conducted in [33] to assess users’ understanding of security features in software applications, only 35% of 340 participants know the meaning of ActiveX control in Internet Explorer.

### 3.4. Poor Mental Model

Ref. [45] pointed out that a security warning as a form of risk communication must be established from the non-technical mental models and metaphors from the real world. They emphasized in their study that different target audiences (users) should construct different mental models.

Ref. [46] conducted a mental model study utilizing 20 users, i.e., 10 technical or advanced users and 10 beginner users. The study was to examine the user’s comprehension of security warnings. They adopted a mental model to obtain a better understanding of how these 20 users react to the warning considering that the mental model is represented with a set item related to logical thinking and reasoning about how a computer warning works. Based on the evaluation results, they concluded that users often have bad mental models.

### 3.5. Unmotivated towards Heeding Warnings

Ref. [47] utilised an eye-tracker to evaluate the user’s response to security indicators on a browser whilst they were on a secure online business. It is worth noting that participants are aware of the lock icon in the status bar, but they only rarely have the willingness to click on the lock, and thereby they are unable to obtain the most out of the site’s certificate. 

Ref. [48] proposed some principles to enhance users’ security behaviour. Accordingly, it has been highlighted that users are ordinarily unmotivated in making security-related decisions. To make matters worse, the end-users seldom read all relevant information and choose not to care about all the possible consequences of their actions. 

Ref. [49] pointed out that users did not prioritize concerns of security aspects as their main focus. Taking into account that the majority of the users hold online activities (e.g., online shopping, checking email, and online banking) in high regard, it is unlikely for them to look for non-task-related security concerns on purpose. Additionally, the majority of the participants in the study claimed themselves to have an advanced sense of security awareness, but it is interesting to mention that they prefer not to check the extended validation (EV) certificate interface that is available in the browser since this is not their primary goal when browsing.

### 3.6. Low Assessment of the Implication of Warnings

Ref. [50] carried out a survey sampling study involving over 6000 Chrome and Firefox users to investigate whether they adhere to real warnings. It has been concluded that site reputation is a vital factor in most users’ decision process and comprehension of the warnings. Participants are confident that they understand the warnings, and they are willing to take the risk and proceed.

### 3.7. Low Evaluation of Risk from Warnings

Ref. [51] conducted a study on certificate warnings to identify factors that may cause the insecure behaviour of users. In the study, participants were presented with various certificate warnings when they sought to access different types of websites and scenarios, i.e., online shopping and banking transactions, social networks, and/or information sites. Based on the results, people who underestimate the actual risk tend to ignore the warnings. It is noteworthy that personal risks (e.g., sleuthing confidential information and/or business loss) were demonstrated to be more effective in minimizing and preventing users from visiting a website than that of general or unknown risk. 

Ref. [52] performed a study to quantify how the warning description text could influence users’ final decisions to comply with pop-up warnings. The results showed that the text description had a useful impact on the time used whist having or assessing the warning. It is also reported that most of the participants who ignored the warning either do not understand the security threat or did not believe they would be at risk if they visited legitimate websites. 

Ref. [53] experimented (100 participants; scenario: online banking transactions) to examine the users’ awareness of the risks and understanding of security warnings. It can be noted that most of the respondents pay much attention to reading and understanding the message content when they respond to the message. On top of this, none of the respondents verified whether the medium of communication was safe by seeking the signal icon, i.e., the lock icon that is available in the address bar and/or the indicator “https”.

### 3.8. Immersion in the Primary Task

Ref. [54] stated that the most efficient way to promote and raise users’ awareness of the security warning is to ensure that the users set the security tasks as their primary goals. It is common that users selectively ignore the security advice to complete their tasks fast, provided the impacts of the security risks are lower than those of the consequences of not completing or delaying the tasks.

Ref. [31] reported that 45% of the respondents tend to ignore the security warning to enable them to concentrate on completing their ‘more’ important tasks. They claimed that they are more than willing to take some risks (by ignoring the security warnings) to complete the job quickly. 

An experiment was carried out to determine whether the end users could distinguish if a pop-up warning was real or fake [55]. The results indicated that most of the respondents reacted to the fake pop-up warning, while nearly half of the participants (42%) claimed that they just wanted to “get rid of the error message” and accomplish their tasks quickly.

### 3.9. Habituation to Security Warning

Ref. [56] studied the effect of habituation on users’ attention maintenance. In this study, attention maintenance could be tracked by identifying the time spent on each of the repeated messages displayed. The results showed that attention maintenance to a message drops rapidly from about 15 s to 2 s with just three exposures to the message. 

Ref. [57] conducted a study to investigate users’ behaviour towards various types of warning messages. It is worth mentioning that the user experience of a warning can have a significant influence on end users’ comprehension of the security warnings’ contexts. Accordingly, it was reported that most of the end users pay less attention to SSL warnings if they are exposed to the warnings frequently.

Ref. [58] claimed that the habituation effect can be significantly bigger if the end users are constantly exposed to the warning’s messages. With this, the effectiveness of a warning might be affected if the users tend to dismiss it. 

Refs. [59,60] also stated that habituation is the primary reason why users ignoring the warnings. The experiment derives from functional magnetic resonance imaging (fMRI) illustrating that the optical or visual treating centres of the brain drop significantly. This occurs only after the warning images are exposed the second time. In other words, this finding implied that habituation will take over the situation, especially after similar images are presented, which significantly affects the human brain.

### 3.10. Summary of Problems and Challenges

Initially, [16] classified seven challenges of security warnings. Then, [20] improved the classifications with eight challenges. Then, our work complimented [16,20]’s classifications by enhancing them accordingly and introducing immersion in the primary task in the problems and challenges as depicted in Table A1. In addition, we present Figure A3 to show the flow and differences in terms of the classification being made. The orange box indicates the new classification group proposed by the authors. It can be noted that we have amended and simplified the classification title, respectively. We also preserve the meaning of each presented classification after the simplification is done. Finally, we introduce the summary as depicted in Table A2 that highlights the updated works and improves the version of the classification of problems and challenges as shown.

It can be noted that the security warning matters have been thoroughly investigated and grouped accordingly by the scholars. Considering these important issues in the computer security warning, this paper continues to probe various approaches towards improving security warnings.

## 4. Approaches to Improve Security Warnings

Based on our assessment, not much work has emphasized classifying the approaches to improving security warnings. Most identified works describe their problems in security warnings independently in publications, e.g., journals, proceedings, or articles. Our work contributes to the body of knowledge within the domain by gathering all the evidence of problems and challenges. Then, we provide the classifications of improvement based on our current assessments. It can be revealed that polymorphic warnings, audited dialogues, interactive design, mental models, attractors, thermal feedback, adaptive security dialogues, facial cues, and alternative security dialogues-Kawaii are the identified methods that have been used as a panacea in warning studies ordinarily. The summary of the mentioned approaches is presented in Table A2, respectively.

### 4.1. Polymorphic

Polymorphic warnings can be considered as an effective solution to improving security warnings, especially in combating habituation [32,59,61,62]. Ref. [32] defined polymorphic dialogues as repeatedly changing the form of warnings that required user inputs. The warnings are designed using context-sensitive guidance (CSG) as a security decision. This main intention is to ensure that users are focusing on security decisions. In their study, they consider two simple dialog alterations. First, the content of the warnings is displayed in a random order. Second, the last option which affirms an alternative will only be activated after the specific dialogues have been displayed within the dedicated amount of time. 

Ref. [59] highlighted that habituation is amongst the grounds of why users ignore warning messages. They determined that there is a gap correlated to habituation that makes it hard to be understood by the users based on the assessment of users’ brain activities. To mitigate the issues, fMRI is used to detect brain activities whilst habituation occurs. They designed new polymorphic warnings utilizing 12 graphical variations to capture users’ attention. 

Ref. [61] embraced a similar approach to [59], where five variations of polymorphic warnings were used. Thirty participants were recruited to experience two types of warnings, namely standard and improved security warnings, i.e., polymorphic. The results show that most participants spend more time on the improved security warning, which indicates that habituation can be reduced. 

On the other hand, ref. [62] reduced the habituation by utilizing the four design variations in the experiment including pictorial symbols, background colour changes, and jiggle and zoom animations. The results indicated that the polymorphic warnings are immune to habituation compared to the standard warning.

### 4.2. Audited Dialog

Audited dialog is known as attempting to hold accountable the truthfulness of one user’s replies or responses. It bilks the wrong answers and warns the users by mentioning that the answers are expected to be submitted for audit purposes, and it will quarantine those who submit undue answers [32]. Researchers implemented context-sensitive guidance (CSG), and the results indicated that most users significantly accept the undue risks, i.e., CSG-polymorphic, compared to the standard dialogues.

### 4.3. Iterative Design

According to [63], iterative design can be defined as a process of development that involves a consistent design stage via user testing and other evaluation methods. Ref. [15] utilized iterative design together with a physical metaphor such as keys, locks, and walls. They compared the Comodo warning (C-warning) with the new warning (P-warning). It can be summarised that most participants opted for the warning design, which communicates better risks and information. 

In addition, another iterative design process was implemented by [64]. They used a five-phase iterative model (ADDIE) that stands for analyse, design, develop, implement, and evaluate. From the experiment utilising eye-tracking, it can be suggested that graphical and interactive components can gauge users’ attention and increase comprehension. Thus, their experiments, i.e., secure comics results, show that their comics can motivate changes in security behaviour. 

Ref. [65] used small focus group workshops to seek out problems and challenges with regard to the current implementation of security warnings. The process was iterated until the fifth workshop. As a result, users’ opinions and perspectives were gathered and contributed directly to the developers or designers improvement of SSL warnings.

### 4.4. Mental Model

A mental model is an inner idea that addresses the operating procedures of one scenario in real life [66]. Ref. [67] added that the mental model can be used to anticipate one person’s conscious mind. 

Ref. [68] investigated the mental models that guided home computer users to make security decisions. From the conducted interviews utilising 33 respondents, he distinguished eight different mental models in two wide groups accordingly: (1) models about viruses, spyware, adware, and other forms of malware under the term of ‘virus’, and (2) models about the attackers, referred to as ‘hackers’. 

Then, Ref. [69] claimed that the mental model in computer security has two main purposes, namely to construct a better efficacious user interface by comprehending the security model of users and as a medium of communication with the users. Ref. [10] acquainted the mental model concept with the differences between advanced and novice users’ perceptions towards security warnings. The mental model of these two groups was then developed and mapped accordingly. It can suggest that warnings should be designed corresponding to the classification of users’ knowledge.

### 4.5. Attractors and Thermal Feedback

Attractors, i.e., icons, words, images, and colours, can be used to attract users’ attention. Ref. [58] proposed the use of attractors to attract users’ attention to an information field (salient field). There are two types of attractors which are inhibitive attractors and non-inhibitive attractors. It can be revealed that users who are exposed to the inhibitive attractors tend to go with an informed decision compared to those in the control condition. 

According to [70], thermal stimulation can be associated with feelings such as emotion and danger. For example, physical danger such as fire can make people recoil their hands from a hot surface. Their study consists of an online questionnaire and lab study to analyse whether a temperature range with different states of web security is associated with people. The results yield that people in general affiliate a cold temperature with a secure page and warm with an insecure page.

### 4.6. Adaptive Security Dialogues

Ref. [71] introduced adaptive security dialogs (ASD) with regard to security-related dialogues. This study aimed to gauge users’ attention when opening a potentially dangerous email attachment. In ASD, some degree of user risks is addressed and correspondingly confirmed in the dialogue’s execution. The adaptation of security warnings dialogs was established from the user’s risk level. It can be revealed that the ASD prototype has a significant improvement when users are rated similarly to all prototypes, which indicates that ASD does not add significant overhead.

### 4.7. Facial Cues

Ref. [72] incorporated facial cues of known menaces or threats into security warnings to attract end users’ attention. With this approach, the facial expression’s validated images that include fright and disgust are integrated into the security warning design. For the fright expressions, it indicates threat that involves physical movement or attack, whereas disgust expressions signal that the environment is contaminated. The facial expressions are utilized to encourage the user’s attention to threats and cultivate a secure manner of behaviour. The results indicate that all activities on the right amygdala are differentially associated with warnings together with facial cues such as disgust, fear, and neutral emotions.

### 4.8. Alternative Security Dialogues-Kawaii

In a recent study by [73], it was observed that users tend to neglect security warning dialogues for two main reasons. The first reason is that the dialogues fail to attract the user’s attention, whereas the second reason is due to users encountering a dialog repeatedly, i.e., habituation. Hence, they propose an alternative implementation of dialog utilising the “Kawaii” effect that can be defined as cute in Japanese. In their experiment, they designed the warnings based on two policies as follows:i.Incorporating “Kawaii” effect;ii.Utilising animation and audible stimulus in the security warning dialog.

The results indicate that the suggested dialog gains better user focus compared to the standard dialogs. In addition, their experiments prove that with the “Kawaii” effect, action toward the dialogue would tend to be disregarded, albeit habituation occurs. All the mentioned approaches would have their strength and limitations to improving security warnings [20].

### 4.9. Console Security Feedback or Advice

Ref. [74] conducted a controlled experiment with 53 participants on application programming interface (API)-integrated security advice warnings. It notified the users about the misused API and addressed secure programming hints as a guide. The results revealed that based on the mentioned approach, it managed to improve the code security, where 73% of participants that experienced the security advice significantly fixed their insecure code. 

Two years later, [75] used the same technique, i.e., the security feedback with 25 professional software developers in a focus group activity. Researchers managed to identify useful security information to avoid insecure cryptographic API use in development. The results suggest that security feedback should be transcending tools and flexible enough for software developers with the consideration of their domain and requirements.

## 5. Proposed Timelines of Problems, Challenges, and Approaches to Improving Security Warnings

Based on the summary in Section 3 and Section 4, a timeline of problems, challenges, and approaches to improving security warnings is proposed as shown in Appendix A—Figure A1 and Figure A2, respectively. Appendix A—Figure A1, i.e., highlighted in yellow, is the list of problems and challenges, whilst Appendix A—Figure A2, i.e., highlighted in orange, is the list of approaches to improving security warnings based on the literature gathered from 1999 to 2020. It is worth noting that a good number of works in security warning implementation were produced and can be considered consistent from a year-to-year basis. There is a consistent trend of works being reported that specifically explore the matters of problems and challenges. On the other hand, a small progression can be seen with the approaches of works to improving security warnings. These timelines are expected to become the reference and avenue for other researchers to comprehend and to analyse the trend that has been gathered, respectively. Although the identified problems and challenges have not been mapped directly to the respective solutions, we believe that the timeline will be useful for providing substance concerning the background or the literature in the security warnings domain.

It is worth noting that the aspects of usable security are becoming more important and more likely to be highlighted because they involve human intervention, especially when people must make a choice and a decision [76,77]. This supports the claim by [78], in which the author stated that “people are the weakest link”. 

Apparently, to the best knowledge of the authors, no work has presented a timeline on the problems, challenges, and approaches to improving security warnings. Thus, these timelines can be considered as a new contribution within the domain of security warnings. It will be useful to researchers for understanding the groundwork, the continuity, and the evolution of security warnings, respectively.

## 6. Discussion

This paper highlights a revealing insight into two main aspects, namely:i.Problems and challenges in security warnings;ii.Approaches to improving security warnings.

It can be noted that most researchers are focusing on identifying the problems, challenges, and solutions separately. We determined that a lack of focus on the classification aspects of the problems and solutions is discussed antecedently. We believe that the identification and understanding of these aspects are important to designing usable security aspects of security warnings. Many factors can be associated with usable security warnings such as development cost, consistency of usage, graphical user interface design, scalability, adaptability, and simplicity. With the correct and appropriate classifications, it eases the process of understanding the foundation or basis of the problems together with the possible solutions that can be put in place. 

How can the findings benefit the research community? This work bridges the gaps by addressing thorough reviews about security warnings at the beginning. Then, it narrows down by gathering works from the various sources by improving the classifications, respectively. Mapping timelines are produced after that, which can be seen as a practicable scheme that complements the problems and solutions in the security warnings niche area, respectively.

### 6.1. Problems, Challenges, and Approaches to Improving Security Warnings 

We present the flow and differences of security warning classifications as depicted in Appendix A—Figure A3. There are some significant changes from the year 2016 to the recent proposal as highlighted in yellow. Previous work introduces habituation to security warning classification, whilst our work added new classification immersion to the primary task, where the primary goal becomes the leading factor based on the user’s reaction to the security warning. Typically, users tend to ignore the security advice in completing their task if users believe that the consequences of security issues are less severe than the consequences of failing to complete the task.

We can view a continuous positive trend, especially in assessing the end user’s problems, challenges, and solutions in security warnings based on the given timeline in Appendix A—Figure A1 and Figure A2. There are a number of works reported for the last 3 years. On the contrary, works related to the approaches to improving security warnings are quite fair in terms of numbers. Although it is not much expanded on a year-to-year basis, there have still been some works reported consistently for the past 3 years. 

Appendix A—Table A1 presents the improvement’s summary of problems and challenges in security warnings. All related works are identified, and groups are based on the classification accordingly. It is expected that these classifications can be expanded by other researchers. Having said that, we can analyse the development now and then, i.e., from time to time. 

In addition, we also present a summary of approaches to improving security warnings, shown in Appendix A—Table A2. Intrinsically, all nine of these approaches are widely used to improve security warnings. The polymorphic and mental model is the most common approach being used. From the usability perspective, the polymorphic warnings will change the form of warning dialogues based on user input. It is much easier to do this by combining with the signal cues, i.e., icons, images, and sounds, as these attributes are easily comprehended by the end users. On the other hand, the mental model is useful to give an overview of the end user’s thought process from different backgrounds. It provides answers to the question of how novice, intermediate, advanced, and expert users perceive the security warnings. Thus, before the design and implementation stages of security warnings are conducted, the outcome from the mental model is much needed. We also believe that the hybrid approaches can be further explored by combining more than one approach to solve issues in security warnings. Therefore, this opens an opportunity for the researchers to gain appropriate outcomes from the experiments conducted. 

Seeing this evidence indicates that the aspects of usable security are continually growing. The researchers keep exploring to understand in-depth the issues from the end users by providing various types of experiments. As the technology evolves, the attackers will similarly take the opportunity to penetrate the system. With the rise of artificial intelligence, for instance, more challenging issues can be highlighted from the experiments conducted. 

### 6.2. Future Trends

Warnings are used on computers as a form of communication. There are many browsers from various developers such as Chrome, Edge, Firefox, Safari, and Opera. Each of these browsers presents security warnings with their methods. To date, no specific or standard approach has been introduced for the developers to design their warnings. However, each of these developers may introduce their standards and guidelines based on user studies that they conducted. Based on our investigations, these are amongst the notable works related to the guidelines and standards that developers may consider to be used in security warning design as depicted in Appendix A—Table A3. There are five guidelines that have been used by the developers to design security warnings. Each of these guidelines utilised different criteria but serve a similar purpose in ensuring usable security is achieved. Therefore, when particular standards are not relevant, the guidelines serve as advice to consumers or developers to follow. For instance, NEAT and SPRUCE are used by Microsoft. On the other hand, HCI-S, secure interaction design, and guidelines for designing usable security mechanisms are generally used by common designers. 

When designing security warnings, the dialogue box type is the common type of warning being presented to the end users. The dialogue box is used for critical warnings that utilise information and when an instant decision is needed from the users. There is some form of works investigating the need to automate the warnings so that the end users’ burden can be alleviated, i.e., the computer will determine the decision on the behalf of the users. However, more evidence and empirical works are needed to support the approach. In addition, some significant improvements have also been made by developers from time to time to improve usable security warnings, e.g., updated browsers, introducing the guidelines, etc. 

## 7. Conclusions

This paper is determined to provide two outcomes. Firstly, it has gathered evidence on problems, challenges, and approaches to improving security warnings. This work complements the previous work; highlights a new classification, i.e., immersion in the primary task; and updates the list, respectively. Secondly, two timelines are proposed to highlight all related works in the security warnings niche area regarding the problems, challenges, and solutions gathered from 1999 to the recent year of 2020. This work also reveals the opportunity for other scholars within the domain to expand our work by introducing possible new classifications. In addition to this, more recent studies can be added to the given timeline. For future work, we plan to map and tailor the problems and challenges with specific solutions to improve security warnings. We believe that the outcome of this research can have a significant impact as a guide and reference for the betterment of security warnings in the future.

## Data Availability

This is a review paper where the authors gather as much as possible evidence from the previous studies related to security warnings and their development.

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
