# Peer review of "Harnessing the Challenges and Solutions to Improve Security Warnings: A Review"

_sensors, 2021, doi:10.3390/s21217313_

Round 1

Reviewer 1 Report

 I think this paper discusses an interesting topic, but some issues need to be solved:

1  Most of your sections are just listed the main idea of papers in references, but lack of comparision, discussion, and re-organization. More discuss needs to be carried out to improve the value of the paper

2 Better give table 3 to demonstrate your view or proposal on the solving of problems or challenges

3  Section 6 needs to be detailed given, currently it is too simple

Author Response

Thank you to the reviewer for all the given feedback. It is very much useful for us to improve the quality of the manuscript. Please find enclosed of the corrections being made based on the reviewer suggestions. We really value your time and effort.

Reviewer Feedback

Author’s Feedback/Corrections

Page Number

1  Most of your sections are just listed the main idea of papers in references, but lack of comparison, discussion, and re-organization. More discuss needs to be carried out to improve the value of the paper

We have improved the manuscript as follows:

-          Add 2 sentences in introduction to signifies the gaps.

-          We introduce Figure A3 to highlight the flow and differences in terms of classification from the various authors (i.e., until the recent one).

-          We extend more in-depth discussion by introducing 2 sub sections namely section 6.1 and 6.2. We discuss further about the outcome from the presented table/figure and future trends.

-          Instead of having 1 timeline that comprises all in original manuscript, we  propose to make it separately (problem and challenges vs approaches to improve) as shown in Figure A1 & A2. This is suggested by the other reviewer for clarity purposes.

-          We introduce Table A3 on works that related to the guidelines and standard that suit with usable security context.

Page 2

Page 5 & Page 13

Page 9 & Page 10

Page 12

Page 10 & Page 16

2 Better give table 3 to demonstrate your view or proposal on the solving of problems or challenges

More explanations are added as suggested from the presented Table A1.

Page 9

3  Section 6 needs to be detailed given, currently it is too simple

We prolong more in-depth discussion by introducing 2 sub sections 6.1 and 6.2. We discuss further about the outcome from the presented table/figure and future trends. We introduce Guidelines for warning design to further support the discussion.

Page 9, Page 10 & Page 16

 Note: All newly added write up is highlighted in yellow color.

Reviewer 2 Report

The manuscript presents a survey on security warnings. From a historical point of view, the representation of the evolution is complete but, despite this, the usefulness for the reader is marginal due to a pair of significant deficiencies.

The first of these is the absence of a section on possible future trends. This type of information is not evident in the work but would be the most useful. Therefore, a short section dealing with this topic is recommended.

The other is the proposed timeline which for the authors is a strength of the work, but as it has been represented it is not of immediate use since it is organized by authors and not by proposed approach. I suggest instead of the authors to modify figure 1 in the proposed direction or at least propose both the two keys of reading in two separate figures.

Author Response

Thank you to the reviewer for all the given feedback. It is very much useful for us to improve the quality of the manuscript. Please find enclosed of the corrections being made based on the reviewer suggestions. We really value your time and effort.

Reviewer Feedback

Author’s Feedback/Corrections

Page Number

The first of these is the absence of a section on possible future trends. This type of information is not evident in the work but would be the most useful. Therefore, a short section dealing with this topic is recommended.

We have improved the manuscript as follows:

We extend more in-depth discussion by introducing 2 sub sections namely section 6.1 and 6.2. We discuss further about the outcome from the presented table/figure and future trends as recommended by the reviewer.

Page 9 & Page 10

The other is the proposed timeline which for the authors is a strength of the work, but as it has been represented it is not of immediate use since it is organized by authors and not by proposed approach. I suggest instead of the authors to modify figure 1 in the proposed direction or at least propose both the two keys of reading in two separate figures.

Instead of having 1 timeline that comprises all in original manuscript, we follow the recommendation by the reviewer to make it separately (problem and challenges vs approaches to improve) as shown in Figure A1 & A2.

Page 12

Additional information

We introduce Table A3 on works that related to the guidelines and standard that suit with usable security context.

Page 10 & Page 16

 Note: All newly added write up is highlighted in yellow color.

Round 2

Reviewer 1 Report

I think the author has solved my issues and now it is OK except some language problems.

Author Response

To reviewer,

Thank you for the feedback. We have improved the paper relating to the language matters as suggested. The usage of words, spelling and grammar have been thoroughly checked. We hope that it satisfies and fulfill the requirements. 

Thank you. 

Reviewer 2 Report

The current version of the manuscript has currently addressed all my suggestions.

The paper is improved and can be considered for a possible publication on Sensors.

Author Response

(The authors gave the same response as above.)
